Journal of Data-centric Machine Learning Research (2024)          Submitted 7/24; Revised 8/24; Published 9/24

# Benchmarking Edge Regression on Temporal Networks

**Muberra Ozmen**                                                        MUBERRA@BLOCK.XYZ
*Cash App*
*Montreal, QC, Canada*

**Florence Regol**                                                       FLORENCER@BLOCK.XYZ
*Cash App*
*Montreal, QC, Canada*

**Thomas Markovich**                                                    TMARKOVICH@BLOCK.XYZ
*Cash App*
*Cambridge, MA, USA*

**Reviewed on OpenReview:** *openreview. net/ forum? id=4k4cocpuSw*

**Editor:** Yue Zhao

## Abstract

Benchmark datasets and task definitions in temporal graph learning are limited to dynamic node classification and future link prediction. In this paper, we consider the task of edge regression on temporal graphs, where the data is constructed from sequence of interactions between entities. Upon investigating graph benchmarking platforms, we observed that the existing open source datasets do not provide the necessary information to construct temporal edge regression tasks. To address this gap, we propose four datasets that naturally lend themselves to meaningful temporal edge regression tasks. We evaluate the performance of a set of method based on popular graph learning algorithms in addition to simple baselines such as vertex-based moving average. Processed versions of proposed datasets are accessible through this repository [1].

**Keywords:**   Temporal Edge Regression, Graph Representation Learning, Edge-wise Graph Learning, Temporal Graph Learning

## 1 Introduction

Graphs are mathematical structures that naturally model the complex and often correlated relationships between entities by modeling the entities as vertices and relationships as edges. This formulation enables modeling approaches that capture the correlated nature of complex systems such as social networks (El-Kishky et al., 2022; Wu et al., 2022; Gao et al., 2021) or financial networks (Liu et al., 2020; Zhang et al., 2022). Recent years have seen the profileration of neural networks for a variety of different graph structured problems (Zhou et al., 2020; Wu et al., 2020; Hamilton, 2020) including fraud detection (Liu et al., 2020; Zhang et al., 2022), recommendation systems (Wu et al., 2022; Gao et al., 2021), chemistry and materials science (Pezzicoli et al., 2022; Reiser et al., 2022; Bongini et al., 2021; Han et al., 2021; Xiong et al., 2021), traffic modeling (Rusek et al., 2019; Chen et al., 2022), and

---

1. `huggingface.co/cash-app-inc`

weather simulation (Keisler, 2022; Ma et al., 2022). Much of this work focused on making predictions about vertices or graphs, where the graph is often assumed to be static.

Acknowledging that most graphs in industrial settings evolve through time, the graph learning community developed a subfield called Temporal Graph Learning (TGL), which focuses on solving tasks such as Dynamic Node Classification (DNC) (Kumar et al., 2019), Future Link Prediction (FLP) (Arnoux et al., 2017), and Recent Link Classification (RLC) (Ozmen and Markovich, 2024). DNC (FLP) can be understood as traditional semi-supervised node classification (link prediction) in a setting where the graph connectivity, graph features, or both can change through time. Both of these tasks have a variety of applications, including predicting the probability of two people forming a friendship, or a person churning their account, corresponding to FLP and DNC tasks respectively (Min et al., 2021; Song et al., 2021; Frasca et al., 2020; Zhang et al., 2021). State of the art methods for both DNC and FLP focus on learning representations for vertices, and ignore many of the details of the edges, aside from the connectivity that they indicate.

By contrast, many problems in industrial settings are concerned with identifying properties about individual edges. Acknowledging this fact, RLC was developed to predict the class of an edge, conditioned on that edge's existence. Industrially relevant applications of this task include identifying messages that are likely to be abusive, or transactions that are likely to be fraudulent. It was found that traditional temporal graph learning methods struggled in RLC settings (Ozmen and Markovich, 2024) when the appropriate modifications were made to the decoder. This suggests that additional, edge-focused, modeling will be required to succeed on this task. Consequently, this makes RLC an exciting new task for the graph learning community. Similar to RLC, it is common to predict a continuous target value associated with an edge, with industrial applications including predicting the time to message response, and delays in shipping or transit. This therefore begs the quesiton – "In analogy to recent link classification, is it possible to define a temporal edge regression task?" To answer this question, we formulate the task of Temporal Edge Regression (TER), which aims to predict a target value for an edge, based on past observations and time-varying graph connectivity. In this work, we present four novel datasets that have been tailored to TER, as well as a variety of benchmark results for strong heuristic baselines and industry standard temporal and static graph neural networks.

The remainder of this work is organized as follows. In Section 2 we present a review of the literature and related work on the topic. We present a clear elucidation of the problem statement in Section 3. In Section 4 we turn our attention to the four new datasets that we have constructed. We present the baseline results in Section 5, and finally we present our conclusions in Section 6.

## 2 Related Work

**Temporal Graph Learning.** There are two main types of dynamic graph models: discrete-time and continuous-time (Zhou et al., 2022). Traditional static graph representations are insufficient for capturing temporal dynamics. Temporal Graph Learning (TGL) models address this limitation by extending graph-based models to time-varying structures (Kumar et al., 2019; Pareja et al., 2020). TGL models can be categorized into two groups: 'time-and-graph' and 'time-then-graph' (Gao and Ribeiro, 2022). Time-and-graph learn

node representations for each graph snapshot, while time-then-graph models construct a multi-graph using past observations and learn node representations. The most well-known TGL method is the Temporal Graph Network (TGN) (Rossi et al., 2020). This is a message passing based encoder which learns graph node embeddings on a continuous-time dynamic multi-graph represented as a sequence of time-stamped events. At time $t$, a memory vector $\mathbf{s}_i(t)$ is stored for each node $i \in \mathcal{V}$. This vector represents the node's history in a compressed format. TGN involves four main operations: (1) message function calculation; (2) message aggregation; (3) memory update; and (4) embedding calculation.

1. **Message Function.** The memory of a node is updated only when it is involved in an event. For example, consider an interaction $\mathbf{e}_{i,j}$ between nodes $i$ and $j$ at time $t$. A message is calculated for each node:

$$\mathbf{m}_i(t) = f_{\mathrm{msg}}\left(\mathbf{s}_i\left(t^-\right), \mathbf{s}_j\left(t^-\right), \Delta t, \mathbf{e}_{i,j}(t)\right), \tag{1}$$

$$\mathbf{m}_i(t) = f_{\mathrm{msg}}\left(\mathbf{s}_i\left(t^-\right), \mathbf{s}_j\left(t^-\right), \Delta t, \mathbf{e}_{i,j}(t)\right). \tag{2}$$

Here $f_{\mathrm{msg}}(\cdot)$ is a learnable function, $\mathbf{s}_i\left(t^-\right)$ denotes the memory of node $i$ just before time $t$, and $\Delta t$ denotes the time since the last interaction. In practice, the inputs are concatenated and a Multi-layer Perceptron is used as the message function $f_{\mathrm{msg}}$.

2. **Message Aggregator.** If there are multiple messages $\mathbf{m}_i(t_1), \ldots, \mathbf{m}_i(t_b)$ involving a node $i$ such that $t_1, \ldots, t_b < t$, they are aggregated using a non-parametric aggregation function $f_{\mathrm{agg}}(\cdot)$:

$$\overline{\mathbf{m}}_i(t) = f_{\mathrm{agg}}\left(\mathbf{m}_i(t_1), \ldots, \mathbf{m}_i(t_b)\right). \tag{3}$$

Candidate aggregation operations include a mean aggregator, i.e., taking the average of all relevant messages, and a 'last' aggregator, i.e., using only the most recent message.

3. **Memory Update.** An autoregressive module, updates the memory state of each node based on the aggregated messages and the current memory of the node:

$$\mathbf{s}_i(t) = f_{\mathrm{mem}}\left(\overline{\mathbf{m}}_i(t), \mathbf{s}_i\left(t^-\right)\right), \tag{4}$$

where $f_{\mathrm{mem}}(\cdot)$ is memory update function.

4. **Embeddings.** The node embeddings are calculated as a function of memory states and events. The embedding module can be as simple as the identity function, which is equivalent to using the memory states directly. Alternatively, in a more complex architecture, multi-head attention can be employed.

TGAT (Xu et al., 2020) incorporates self-attention and time encoding to predictively generate embeddings for new and existing nodes. CAWN (Wang et al., 2021) utilizes temporal random walks for inductive representation of temporal networks. Graph Mixer (Cong et al., 2023) is a simple model that achieves state-of-the-art performance on benchmark tasks. In general, existing TGL methods are benchmarked only on Dynamic Node Classification (DNC) and Future Link Prediction (FLP) tasks (Poursafaei et al., 2022; Huang et al., 2023). More recently, Liu et al. (2024) introduced a framework Temporal Graph

Clustering (TGC) that extends node clustering task widely investigated on static graphs to temporal settings. The proposed method that combine a temporal module for time-based data extraction with a clustering module for node grouping. This integration leverages deep clustering techniques tailored for interaction sequence-based batch processing in temporal graphs. The authors also enhance existing models with clustering assignment distribution and adjacency matrix reconstruction, expanding the scope of deep graph clustering to temporal graphs comprehensively. Despite the growing interest in temporal graph learning, a significant research gap remains in edge-wise learning on temporal graphs (Chanpuriya et al., 2023; Wang et al., 2023; Suresh et al., 2023), particularly when the target variable is continuous. This is largely due to the lack of a suitable benchmark dataset that aligns with this specific problem setting, hindering the exploration of this important area. We refer the reader to Feng et al. (2024) for a comprehensive and recent review of temporal graph learning methods.

**Existing Benchmark Datasets.** We investigate four existing temporal graph learning benchmarking platforms:

- *PyTorch Geometric* [2] provides `Jodie`, a collection of temporal graph datasets for predicting dynamic embedding trajectories, `EllipticBitcoinTemporalDataset`, a Bitcoin transaction dataset labeled as licit or illicit, `GDELT`, a dataset of events collected from 2018 to 2020, and `GDELTLite`, a reduced version of GDELT with events from 2016 to 2020.

- *PyTorch Geometric Temporal* [3] offers several datasets, including `ChickenpoxDataset`, a county-level chickenpox cases dataset from Hungary, `PedalMe`, a PedalMe Bicycle deliver orders dataset from London, `WikiMaths`, a vital mathematics articles dataset from Wikipedia, and three `WindmillOutput` datasets for hourly energy output of windmills from a European country. Additionally, there are `METRLA` and `PemsBay` for traffic forecasting, `EnglandCovid` for mobility and COVID-19 cases in England, `MontevideoBus` for inflow passenger data at bus stop level from Montevideo city, `TwitterTennis` for Twitter mention graphs related to major tennis tournaments.

- *SNAP* [4] provides various datasets, including `soc-RedditHyperlinks` for hyperlinks between subreddits on Reddit, `sx-stackoverflow` for comments, questions, and answers on Stack Overflow, `wiki-talk-temporal` for users editing talk pages on Wikipedia, `email-Eu-core-temporal` for e-mails between users at a research institution, `CollegeMsg` for messages on a Facebook-like platform at UC-Irvine, `soc-sign-bitcoin-otc` for Bitcoin OTC web of trust network, `act-mooc` for student actions on a MOOC platform with student drop-out labels, and `comm-f2f-Resistance` for dynamic face-to-face interaction network between groups of people.

- *TGB* [5] offers several datasets, including `tgbl-wiki-v2` for co-editing network on Wikipedia pages, `tgbl-review-v2` for Amazon product review network, `tgbl-coin`

---

2. `pytorch-geometric.readthedocs.io/en`
3. `pytorch-geometric-temporal.readthedocs.io/en`
4. `snap.stanford.edu/data`
5. `tgb.complexdatalab.com`

for cryptocurrency transaction dataset, `tgbl-comment` for Reddit reply network, and `tgbl-flight` for international flight network.

Unfortunately, our investigation into various datasets failed to yield any that possess the necessary properties for a temporal edge regression task.

## 3 Problem Definition

We are considering a dynamic graph setting in which the set of vertices is fixed, and the edges are appearing over time. We are given a set of vertices $\mathcal{V} = \{u_i\}_{i=1}^N$, and each vertex $u_i \in \mathcal{V}$ is endowed with a $d^v$-dimensional feature vector $\mathbf{x}_i \in \mathbb{R}^{d^v}$. On this system of vertices, we observe pairwise interactions over time represented as edges. An edge $e_j$ is composed of 4 components; the index of the source vertex $s_j$, the index of the destination vertex $d_j$, a $d^e$-dimensional feature vector of the edge $\mathbf{z}_j \in \mathbb{R}^{d^e}$, and the target value of interest $y_j \in \mathbb{R}$. Each of these components can be observed at different times, so we additionally define the time of observation;

- $t_j^x$: time of observation of the source and destination vertex indices,

- $t_j^z$: time of observation of the edge features,

- $t_j^y$: time of observation of the edge target value,

with the constraint that $t_j^x \leq t_j^z \leq t_j^y$. That is, $t_j^x$ marks the announcement time of interaction $j$ and $t_j^y$ marks the completion time of the interaction. The complete information of an edge is given by;

$$e_j = (s_j, d_j, \mathbf{z}_j, y_j, t_j^x, t_j^z, t_j^y). \tag{5}$$

At any time $t$, we can therefore define a set of completed interactions $\mathcal{E}^{\text{completed}}(t) = \{j : t_j^y \leq t\}$ and a set of announced but incomplete interactions $\mathcal{E}^{\text{announced}}(t) = \{j : t_j^x \leq t, t_j^y > t\}$. In Temporal Edge Regression (TER), we aim to learn a classifier that maps the "known" of announced interactions $\mathcal{E}^{\text{announced}}(t)$ to their target value, given the set of completed interactions $\mathcal{E}^{\text{completed}}(t)$ at any time $t$. We further define variations of the problem based on the timing of observation. If the system imposes the following sequence of events:

$$t_j^x = t_j^z < t_j^y,$$

the problem is defined as **Recent Link Regression (RLR)**. That is, at the inference time the source and destination vertices and edge features are known. If the system imposes the following sequence of events:

$$t_j^x < t_j^z = t_j^y,$$

the problem is defined as **Proximate Link Regression (PLR)**. That is, at the inference time the source and destination vertices are known. If the system imposes the following sequence of events:

$$t_j^x < t_j^z < t_j^y,$$

the problem is defined as **Future Link Regression (FLR)**. That is, at the inference time nothing is known about the future interactions.

Table 1: Dataset Statistics

|  | epic-games-plr | air-traffic-2019-rlr | air-traffic-2015-rlr | open-sea-rlr |
|---|---|---|---|---|
| # of src | 542 | 274 | 257 | 1,932,463 |
| # of dst | 614 | 274 | 257 | 1,758,601 |
| # of nodes | 1,156 | 274 | 257 | 2,601,107 |
| # of edges | 17,584 | 484,551 | 5,138,263 | 25,876,360 |
| # of timestamps | 3,267 | 181 | 334 | 7,361,184 |
| # of node features | 573 | 0 | 0 | 0 |
| # of edge features | 512 | 20 | 20 | 86 |
| average node degree | 15.21 | 1,768.43 | 19,993.24 | 9.95 |
| average # of repetitions | 1.03 | 143.49 | 1,166.73 | 1.48 |
| maximum # of repetitions | 6 | 1,920 | 13,406 | 10,583 |

## 4 Datasets

This paper introduces four novel datasets: `epic-games-plr`, `air-traffic-2015-rlr`, `air-traffic-2019-rlr`, and `open-sea-rlr`. The `epic-games-plr` dataset involves predicting critic ratings, which are only observable after the critic's identity is revealed, thereby categorizing it as a Proximate Link Regression (PLR) problem. In contrast, the `air-traffic-2015-rlr` and `air-traffic-2019-rlr` datasets aim to predict flight delays based on pre-determined flight characteristics and weather conditions, falling under the category of Recent Link Regression (RLR). Similarly, the `open-sea-rlr` dataset targets predicting the profitability of NFT transactions, where features are observed simultaneously, but profits are only known after subsequent trades, also classified as RLR. Future Link Regression (FLR) cases are left for future exploration. This section provides an in-depth examination of each dataset, with statistical summaries presented in Table 1.

### 4.1 Epic Games

**Description.** Epic Games Store [6] is a digital video game storefront. de Souza Gomes (2022) provides a dataset that contains information on the games released on the platform and their critics provided by different resources. Relevant to our work, the dataset includes two types of records: *game* and *critic*.

**Graph Construction.** The critic records are used to define the graph $\mathcal{G}(\mathcal{V}, \mathcal{E})$ such that the source and destination of the critiques, i.e., the authors' companies and game identities, form the set of vertices and each critic denotes a temporal edge between them. Each vertex, i.e., an author company or game, $u_i \in \mathcal{V}$ is associated with a feature vector $\mathbf{x}_i \in \mathbb{R}^{d^v}$ where $d^v$ denote the feature dimensionality. The features of game vertices are extracted by textual data such as game description, nominal data such as genres, and interval data such as price. An edge $(s_j, d_j) \in \mathcal{E}$, i.e., a critic released by company $s_j$ on game $d_j$, is associated with timestamp $t_j$ which and rating score $y_j \in \mathbb{R}$ which denotes the overall rating provided by the author. The joint density of edge target $y_j$ and time $t_j$ is visualized in Figure 1 along with the degree centrality distribution of the vertices. The preprocessing details followed to construct the graph, calculate the raw feature vectors and response variable are provided in Appendix A.1.

---

6. `store.epicgames.com`

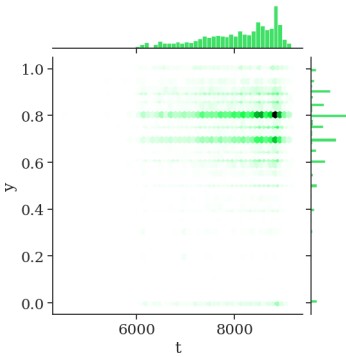 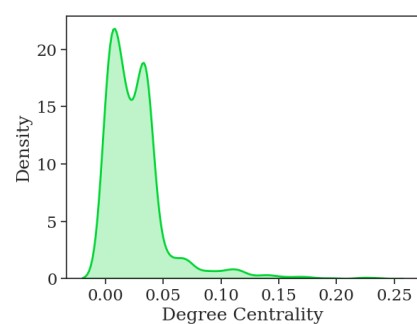

(a) The joint density of edge target and time reveals a noticeable shift in distribution along the time axis. This observation suggests that the time components play a significant role in target prediction, for that dataset.

(b) Vertex degree distribution. This indicates that the dynamic graphs are highly sparse, with most of vertices having few or no neighbors.

Figure 1: `epic-games-plr` graph statistics

**Sequence of Events.** The events associated with an interaction $(s_j, d_j) \in \mathcal{E}$ are observed in the following order:

1. The identity of source node $s_j$ and destination node $d_j$ are observed along with the node features $\mathbf{x}_j \in \mathbb{R}^{d^v}$.
2. The source node interacts with the destination node by releasing a review with an overall rating, which is observed as $y_j \in \mathbb{R}$.

This is fitting because a game will be given to a known set of reviewers, and put under embargo while the reviewer plays the game and considers their review. Therefore, the edge is known ahead of time, and we seek to regress on the future review score.

### 4.2 Air Traffic

**Description.** The Bureau of Transportation Statistics, under the United States Department of Transportation, monitors and reports on the on-time performance of domestic flights operated by major airlines. The datasets for 2019 (Trivedi, 2021) and 2015 (of Transportation, 2017) are publicly available on Kaggle [7] to enable analyses of flight delays and airport performance. Each dataset consists of flight records for to the corresponding year. The flight records contain information on source and destination airports and scheduled and actual departure and arrival times. To investigate the impact of weather conditions on flight delays, we have supplemented the flight datasets with weather data from Open-Meteo [8], an open-source weather API (Zippenfenig, 2023). We extracted weather conditions at the scheduled departure times for both origin and destination airports, enabling the analysis of weather-related delay predictions. This integration enhances the datasets' capabilities, allowing for a more comprehensive understanding of weather's role in flight delay dynamics.

---

7. `kaggle.com`
8. `open-meteo.com/en/docs`

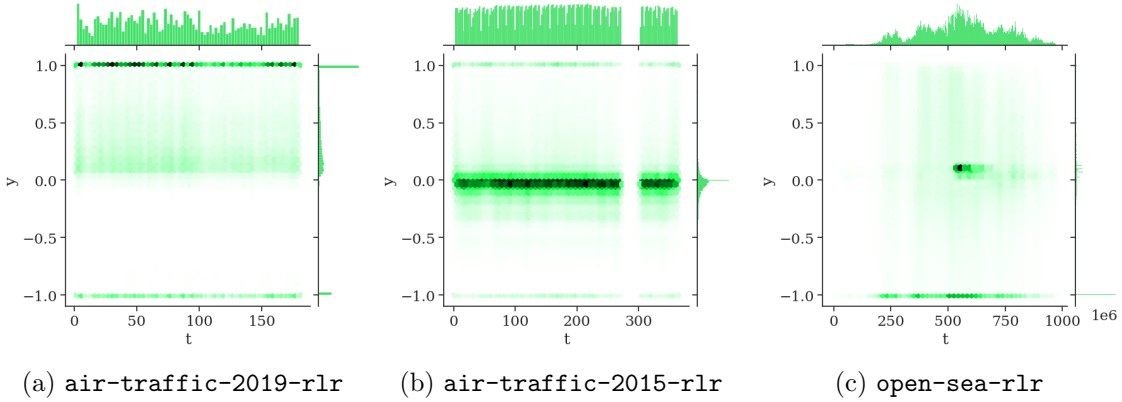

(a) `air-traffic-2019-rlr`  (b) `air-traffic-2015-rlr`  (c) `open-sea-rlr`

Figure 2: Joint density of edge target and time. In both the `air-traffic-2015-rlr` and `air-traffic-2019-rlr`, the distribution of target values remains relatively constant over time, with potentially some periodicity. However, in the OpenSea dataset, the distribution exhibits high irregularity across time, resembling the pattern observed in the `epic-games-plr` dataset.

Notably, our research marks the first exploration of this dataset in the context of graph learning, offering a novel perspective on its applications.

**Graph Construction.** We construct a graph $\mathcal{G}(\mathcal{V}, \mathcal{E})$ where airports are represented as vertices and flights as edges. The vertices lack feature attributes. Each edge $(s_j, d_j) \in \mathcal{E}$, symbolizing a flight from airport $s_j$ to $d_j$, is associated with a timestamp $t_j$ indicating the flight date and a feature vector $\mathbf{z}_j$ derived from weather conditions at the endpoint airports, comprising: daily precipitation sum, maximum and minimum daily air temperature, maximum wind speed and gusts on the flight day. Edge targets $y_j \in \mathbb{R}$, defined as arrival delay normalized by flight duration, representing the delay outcome. The preprocessing steps followed to construct the graph, calculate the raw feature vectors and response variable are further detailed in A.2.

**Sequence of Events.** For a flight $(s_j, d_j) \in \mathcal{E}$ the events follow an order as follows:

1. At the time of take off, the information regarding the scheduled departure and arrival times are known in addition to the weather conditions at the end points.
2. The actual delay $y_j \in \mathbb{R}$ is observed only once the flight is landed.

### 4.3 Open Sea

**Description.** Open Sea [9], a prominent trading platform in the Web3 ecosystem, is the source of a dataset comprising Non-Fungible Token (NFT) transactions. Initially introduced by La Cava et al. (2023); Cava et al. (2023); Costa et al. (2023), this dataset is a collection of Non-Fungible Token (NFT) transactions. Sourced from Open Sea, it is provided as a natural language processing dataset and is mainly used for multimodal learning classification tasks.

---

9. `opensea.io`

**Graph Construction.** We define a graph $\mathcal{G}(\mathcal{V}, \mathcal{E})$ where vertices represent unique sellers and buyers of NFTs, identified by collection memberships and token IDs. Each transaction $(s_j, d_j) \in \mathcal{E}$ is associated with a timestamp $t_j$ and a feature vector $\mathbf{z}_j \in \mathbb{R}^{d_e}$, comprising binary representations of categorical variables, cryptocurrency exchange rates, and monetary values. Notably, the vertices lack raw feature vectors, as the identities of buyers and sellers remain unknown. The target value $y_j \in \mathbb{R}$ of transaction $i$ is calculated as the rate of return on investment, determined by the difference between the revenue from a future sale and the purchase price, normalized by the original purchase price. This metric reflects the profitability of each transaction. The preprocessing steps followed to construct the graph, calculate the raw feature vectors and response variable are further detailed in A.3.

**Sequence of Events.** The events associated with an interaction $(s_j, d_j) \in \mathcal{E}$ unfold in the following sequence:

1. The identities of the source node $s_j$ and destination node $d_j$ are observed, accompanied by the edge features $\mathbf{z}_j \in \mathbb{R}^{d_e}$.
2. The profit of current transaction $y_j \in \mathbb{R}$ is only revealed when the associated NFT is traded again, thereby introducing a delay in the observation of edge targets.

The delay on observing edge targets aligns with the Recent Link Regression (RLR) setting, where the outcome of an investment's profitability is unknown at the time of purchase.

## 5 Methodology

In this section, we (1) define edge homophily which measures the correlation of neighbouring edges on edge target variable, (2) share the details of models used to set baselines on temporal edge regression task, (3) formulate the loss functions used to train neural networks for edge regression task.

### 5.1 Edge Homophily

We introduce a measure of edge homophily to understand the importance of graph information for temporal edge regression task. Denote by $\mathcal{N}^e(\alpha)$ an edge-wise neighbourhood operator that constructs the set of all edges that are connected to a given edge, $\alpha = (s_j, d_j)$, where $s_j$ and $d_j$ are the source and the destination. This operator forms the union of two sets, i.e., $\mathcal{N}^e(\alpha) = \mathcal{I}(j) \cup \mathcal{O}(j)$, where $\mathcal{I}(j)$ is the set of outgoing edges connected to the source $s_j$ and $\mathcal{O}(j)$ is the set of incoming edges connected to the destination $d_j$. Edge homophily measure is then defined as:

$$\bar{\mathcal{H}}_e(\mathcal{G}) = \frac{1}{|\mathcal{E}|} \left( \sum_{\alpha \in \mathcal{E}} \left( y_\alpha - \frac{\sum_{\beta \in \mathcal{E}} y_\beta}{|\mathcal{E}|} \right)^2 - \sum_{\alpha \in \mathcal{E}} \left( y_\alpha - \frac{\sum_{\beta \in \mathcal{N}_\alpha^{(e)}} y_\beta}{\left| \mathcal{N}_\alpha^{(e)} \right|} \right)^2 \right) \tag{6}$$

Edge-homophily measures the correlation between the target values of neighbouring edges in analogy to the way node-homophily measures the fraction of neighbouring nodes with the same class. Node-homophily is an important dataset property that can be highly indicative of the value that can be derived by encoding graph structure in node classification tasks (Pei et al., 2020), particularly for embedding procedures that rely on smoothing over

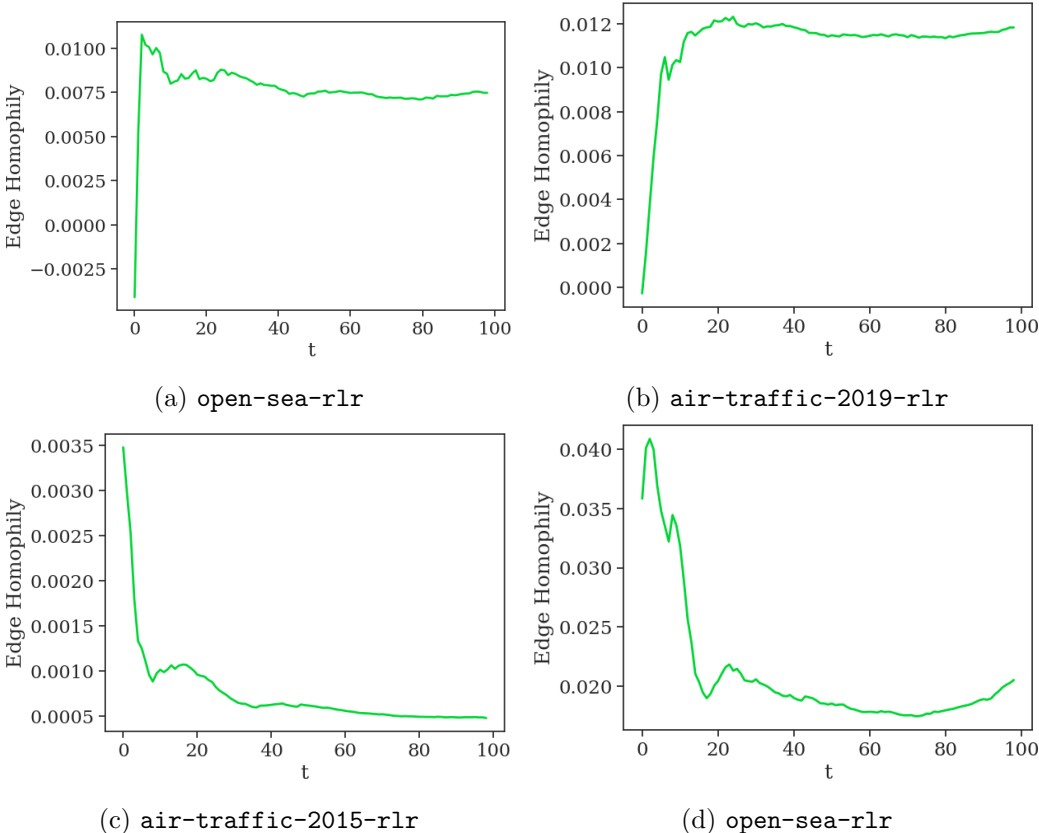

(a) `open-sea-rlr`  (b) `air-traffic-2019-rlr`

(c) `air-traffic-2015-rlr`  (d) `open-sea-rlr`

Figure 3: Edge homophily trends. The x-axis represents the time of observation, while the y-axis represents the edge homophily measure. Generally, after an initial warm-up period, the edge homophily measure stabilizes across all datasets, though magnitude of the measure varies between them.

a neighbourhood. In Figure 3, we illustrate the dynamics of edge homophily over time for all four datasets.

## 5.2 Prediction Methods

In our experiments, we use two non-parametric baselines, a graph-agnostic neural network, four widely used graph learning methods and a temporal graph neural network. In this section, we share the details of each method.

**Non-parametric Baselines.** In order to set baseline, we use two non-learnable prediction methods *Moving Average (MA)* and *Edge Similarity (ES)*.

**Moving Average (MA)** method is a statistical technique used to analyze time series data by smoothing out fluctuations and highlighting trends. It involves calculating the average value of a sequence of data points over a specified period $K$, called the window size. For an announced edge $e_i \in \mathcal{E}^{\text{announced}}$ with source vertex $s_i$ and destination vertex $d_i$, and the set of completed edges $\mathcal{E}^{\text{completed}}(t_i^x)$, we determine sets of most recent edges

inductively. Let $\mathbb{MA}_{\text{all}}(1)$, $\mathbb{MA}_{\text{src}}(1)$ and $\mathbb{MA}_{\text{dst}}(1)$ denote most recent completed edge, most recent completed edge with same source vertex, and most recent completed edge with same destination vertex, respectively. That is, $\mathbb{S}_{\text{all}}(1) = \{e_j : j = \text{argmax}\, t_j^y, e_j \in \mathcal{E}^{\text{completed}}(t_i^x)\}$, $\mathbb{S}_{\text{src}}(1) = \{e_j : j = \text{argmax}_{s_j=s_i}\, t_j^y, e_j \in \mathcal{E}^{\text{completed}}(t_i^x)\}$, $\mathbb{S}_{\text{dst}}(1) = \{e_j : j = \text{argmax}_{d_j=d_i}\, t_j^y, e_j \in \mathcal{E}^{\text{completed}}(t_i^x)\}$. Then we define the $(k+1)^{th}$ order sets as follows:

$$\mathbb{S}_{\text{all}}(k+1) = \{e_j : j = \text{argmax}\, t_j^y, e_j \in \mathcal{E}^{\text{completed}}(t_i^x) \setminus \cup_{r=\{1,\dots,k\}}\mathbb{S}_{\text{all}}(r)\}\}, \qquad (7)$$

$$\mathbb{S}_{\text{src}}(k+1) = \{e_j : j = \text{argmax}\, t_j^y, e_j \in \mathcal{E}^{\text{completed}}(t_i^x) \setminus \cup_{r=\{1,\dots,k\}}\mathbb{S}_{\text{src}}(r)\}\}, \qquad (8)$$

$$\mathbb{S}_{\text{dst}}(k+1) = \{e_j : j = \text{argmax}\, t_j^y, e_j \in \mathcal{E}^{\text{completed}}(t_i^x) \setminus \cup_{r=\{1,\dots,k\}}\mathbb{S}_{\text{dst}}(r)\}\}. \qquad (9)$$

The final sets of most recent edges, most recent edges with same source node, most recent edges with same destination nodes are set by union of iterative sets, i.e., $\mathbb{S}_{\text{all}}(K) = \bigcup_{k=1,\dots,K} \mathbb{S}_{\text{all}}(k)$, $\mathbb{S}_{\text{src}}(K) = \bigcup_{k=1,\dots,K} \mathbb{S}_{\text{src}}(k)$, $\mathbb{S}_{\text{dst}}(K) = \bigcup_{k=1,\dots,K} \mathbb{S}_{\text{dst}}(k)$. Finally, the estimations are made as the average of edges involved in these sets:

$$\textbf{MA-all (K):} \quad \hat{y}_j = \frac{1}{K} \sum_{k \in \mathbb{S}_{\text{all}}(K)} y_k, \qquad (10)$$

$$\textbf{MA-src (K):} \quad \hat{y}_j = \frac{1}{K} \sum_{k \in \mathbb{S}_{\text{src}}(K)} y_k, \qquad (11)$$

$$\textbf{MA-dst (K):} \quad \hat{y}_j = \frac{1}{K} \sum_{k \in \mathbb{S}_{\text{dst}}(K)} y_k. \qquad (12)$$

**Edge Similarity (ES)** is a heuristic averaging model based on edge similarity. The similarity between two edges, $e_j$ and $e_k$, is defined using the cosine distance between their edge feature vectors: $\cos_{j,k} \triangleq \frac{\mathbf{z}_j^\top \cdot \mathbf{z}_k}{||\mathbf{z}_j||\,||\mathbf{z}_k||}$. Given the ordered set of edge indices similar to $e_j$ at time $t$;

$$\text{Sim}(e_j, t) = \Big(k_i;\ i < j \implies \cos_{j,k} > \cos_{i,k}, \quad e_k \in \mathcal{E}^{\text{completed}}(t)\Big), \qquad (13)$$

we return the average of the $K$-th most similar edges to $e_j$ as the prediction:

$$\textbf{ES (K):} \quad \hat{y}_j = \frac{1}{K} \sum_{k \in \text{Sim}(e_j, t)_{[1:K]}} y_k. \qquad (14)$$

**Non-temporal Neural Networks.** To establish a baseline for our experiments, we present six neural network architectures: **eMLP** based on *Multi-layer Perceptron (MLP)* as a graph-agnostic neural network and four graph neural networks **eGCN, eGraphSage, eGAT, eGraphTransformer** based on the widely used *Graph Convolutional Network (GCN)* (Kipf and Welling, 2017), *Graph Sage* (Hamilton et al., 2017), *Graph Attention Network (GAT)* (Veličković et al., 2018), and *GraphTransformer* (Yun et al., 2019). Each architecture has two variants: **eNN** and **eNN-rich**. The e-version refers to a straightforward implementation of the original method combined with an edge predictor module. The

Table 2: Summary of the inputs to the various modules for our different baseline models and their rich variants to obtain the prediction $\hat{y}_j$ of the edge $e_j$. The subscript $s_j$ and $d_j$ denote the node indices of the source and destination of edge $e_j$, while $t_j^x$ is the timestamp of edge $j$ announcement. $\sigma$ is a readout function such as sigmoid or hyperbolic tangent, $f_{\text{out}}$ is one layer of linear transformation, and $||$ denotes concatenation.

| Model | $f_{\text{conv}}(\cdot)$ | $f_{\text{out}}(\cdot)$ | $f_{\text{out}}\text{-}\mathbf{rich}(\cdot)$ |
|---|---|---|---|
| **eMLP** | na | $\hat{y}_j = \sigma\left(f_{\text{out}}\left(\mathbf{h}_j^e\right)\right)$ | $\hat{y}_j = \sigma\left(f_{\text{out}}\left(\mathbf{h}_j^e||\mathbf{h}_{t_j^x}^t\right)\right)$ |
| **eGCN/eGSage** | $f_{\text{conv}}(\mathbf{X}, \mathcal{G})$ | $\hat{y}_j = \sigma\left(f_{\text{out}}\left(\mathbf{h}_{s_j}^v||\mathbf{h}_{d_j}^v\right)\right)$ | $\hat{y}_j = \sigma\left(f_{\text{out}}\left(\mathbf{h}_{s_j}^v||\mathbf{h}_{d_j}^v||\mathbf{h}_j^e||\mathbf{h}_{t_j^x}^t\right)\right)$ |
| **eGat/eGTransf.** | $f_{\text{conv}}(\mathbf{X}, \mathcal{G}, \mathbf{Z})$ | $\hat{y}_j = \sigma\left(f_{\text{out}}\left(\mathbf{h}_{s_j}^v||\mathbf{h}_{d_j}^v\right)\right)$ | $\hat{y}_j = \sigma\left(f_{\text{out}}\left(\mathbf{h}_{s_j}^v||\mathbf{h}_{d_j}^v||\mathbf{h}_{t_j^x}^t\right)\right)$ |

rich version incorporates additional data components that were not utilized by the original method through the edge predictor module. In our experiments, we keep the number of learnable parameters equal on all variants to make a fair comparison. There are three key components that differ between these architectures:

**Node embeddings** calculation is achieved by convolution operation in graph learning methods. Specifically, convolutional layers take node features and graph adjacency matrices as input and generate a set of node embeddings. GCN employs a normalized adjacency matrix to compute node embeddings. GraphSage can be seen as a variant of GCN, where neighborhood aggregation is sampling-based and the aggregation function is more generalized. GAT and Graph Transformer incorporate attention mechanisms, which enable them to combine edge features and node features to generate node embeddings. Let $f_{\text{conv}}(\cdot)$ denote node embeddings module; its output can be summarized as an embedding vector $\mathbf{h}_i^v \in \mathbb{R}^d$ for each vertex $u_i \in \mathcal{V}$, where $d$ denotes model dimensionality.

**Edge embeddings** are generated using a simple two-layer neural network with ReLU activation, which takes edge features $\mathbf{z}_j$ as input. Let $f_{\text{mlp}}(\cdot)$ denote edge feature encoder, its output can be summarized as an edge embedding vector $\mathbf{h}_j^e \in \mathbb{R}^d$ for each edge $e_j \in \mathcal{E}$.

**Time embeddings** are used to capture the temporal component of the interactions, enabling the model to understand and predict temporal patterns and changes within the graph data. For the time encoder, we employ learnable time projection introduced by Kumar et al. (2019). Let $f_{\text{time}}(\cdot)$ denote the time encoder, its output can be summarized as a time embedding vector $\mathbf{h}_j^t \in \mathbb{R}^d$ for a timestamp $t \in \mathbb{R}$.

In **eMLP**, only edge attribute transformation is used to encode a given edge. In **eMLP-rich**, we additionally concat time encoding. In **eGCN** and **eGraphSage**, we encode only the node embeddings based on node features, in **eGCN-rich** and **eGraphSage-rich** we also encode edge attributes and time encoding. In **eGAT** and **eGraphTransformer**, the node embeddings are calculated from the node and edge features, so in **eGAT-rich** and **eGraphTransformer-rich** we only add time encoding. Table 2 provides a summary of the input and final decoder for each method.

**Temporal Graph Network (TGN).** The most well-known temporal graph learning method is the Temporal Graph Network (TGN) (Rossi et al., 2020). This is a message

Table 3: Model Comparison Results. Best and second-best performance are highlighted in red and blue, respectively.

| | epic-games-plr | | air-traffic-2019-rlr | | air-traffic-2015-rlr | | open-sea-rlr | |
| --- | --- | --- | --- | --- | --- | --- | --- | --- |
| | MAE | RMSE | MAE | RMSE | MAE | RMSE | MAE | RMSE |
| **MA-all** (K=10) | 0.1322 | 0.1954 | 0.4510 | 0.5931 | 0.1877 | 0.3005 | 0.3812 | 0.4963 |
| **MA-src** (K=10) | 0.1480 | 0.2499 | 0.4421 | 0.5833 | 0.1864 | 0.2993 | 0.3737 | 0.4910 |
| **MA-dst** (K=10) | 0.2951 | 0.4341 | 0.4444 | 0.5867 | 0.1848 | 0.2970 | 0.3470 | 0.4677 |
| **ES** (K=10) | 0.1374 | 0.2017 | 0.4032 | 0.5515 | 0.1854 | 0.2991 | 0.3870 | 0.4939 |
| **eMLP** | **0.1054** | **0.1755** | 0.4325 | 0.5741 | 0.1727 | 0.2946 | 0.3523 | 0.4707 |
| **eMLP-rich** | 0.1188 | 0.1891 | 0.4282 | 0.5692 | 0.1684 | 0.2971 | 0.3588 | 0.4717 |
| **eGCN** | 0.1178 | 0.1883 | **0.3983** | **0.5467** | 0.1673 | 0.2965 | 0.3551 | **0.4638** |
| **eGCN-rich** | **0.1062** | **0.1788** | 0.4151 | 0.5593 | 0.1727 | 0.2975 | 0.3502 | 0.4693 |
| **eGSage** | 0.1205 | 0.1894 | 0.4021 | 0.5502 | 0.1727 | 0.2975 | 0.3550 | 0.4750 |
| **eGSage-rich** | 0.1191 | 0.1883 | 0.4028 | 0.5516 | **0.1667** | **0.2933** | 0.3520 | 0.4663 |
| **eGAT** | 0.1180 | 0.1883 | 0.4252 | 0.5699 | **0.1665** | 0.2944 | **0.3395** | **0.4573** |
| **eGAT-rich** | 0.1186 | 0.1888 | **0.3995** | **0.5474** | 0.1669 | **0.2942** | 0.3509 | 0.4654 |
| **eGTransf** | 0.1191 | 0.1894 | 0.4174 | 0.5597 | 0.1678 | 0.2964 | 0.3556 | 0.4673 |
| **eGTransf-rich** | 0.1190 | 0.1887 | 0.4028 | 0.5507 | 0.1689 | 0.2974 | **0.3357** | 0.4672 |
| **TGN** | 0.1198 | 0.1888 | 0.4084 | 0.5696 | 0.1689 | 0.2958 | 0.3535 | 0.4648 |

passing based encoder which learns graph node embeddings on a continuous-time dynamic multi-graph represented as a sequence of time-stamped events. At any time, a memory vector is stored for each node. This vector represents the node's history in a compressed format. TGN involves four main operations: (1) message function calculation; (2) message aggregation; (3) memory update; and (4) embedding calculation. In our experiments, we use TGN as a representative memory based neural network.

### 5.3 Loss Functions

Let $\hat{y}_j$ denote the response value predicted by the classifier for $j^{\text{th}}$ edge; the quality of estimation can be evaluated by the mean square error $\mathbb{L}_{\text{mse}}(y_j, \hat{y}_j)$, mean absolute error or $\mathbb{L}_{\text{mae}}(y_j, \hat{y}_j)$ during training:

$$\mathbb{L}_{\text{mse}} = \sum_j \left(y_j - \hat{y}_j\right)^2, \quad \mathbb{L}_{\text{mae}} = \sum_j |y_j - \hat{y}_j| \tag{15}$$

Alternatively, we can use Huber loss:

$$\mathbb{L}_{\text{huber}}(y, \hat{y}) = \begin{cases} 0.5\left(y_j - \hat{y}_j\right)^2, & \text{if } |y_j - \hat{y}_j| < \delta \\ \delta(|y_j - \hat{y}_j| - 0.5\delta) & \text{otherwise} \end{cases} \tag{16}$$

In our experiments, we tune each model individually to select between $\mathbb{L}_{\text{mse}}$, $\mathbb{L}_{\text{mae}}$ and $\mathbb{L}_{\text{huber}}$.

## 6 Experiments

In order to understand TER as a novel temporal graph learning task, we evaluate a range of baselines which have been detailed in Section 5 using Mean Absolute Error (MAE) and

Root Mean Squared Error (RMSE). Let $y_j$ and $\hat{y}_j$ denote actual and estimated target value for edge $e_j \in \mathcal{E}$, and let $\bar{y}$ denote the mean target value, the performance metrics are formulated as follows:

$$\text{MAE} = \frac{1}{|\mathcal{E}|} \sum_j |y_j - \hat{y}_j|, \quad \text{RMSE} = \sqrt{\frac{1}{|\mathcal{E}|} \sum_j (y_j - \hat{y}_j)^2}. \tag{17}$$

**Results.** In Table 3, we present the results for MAE and RMSE. In Figures 4, 5, and 6 we demonstrate the relative performance of methods grouped by their various properties. In Figure 4, we compare non-parametric baselines, graph-agnostic neural networks, static graph neural networks and temporal graph neural network. In Figure 5, we make a comparison within static graph neural networks. In Figure 6, we compare the vanilla and rich versions of deep learning methods.

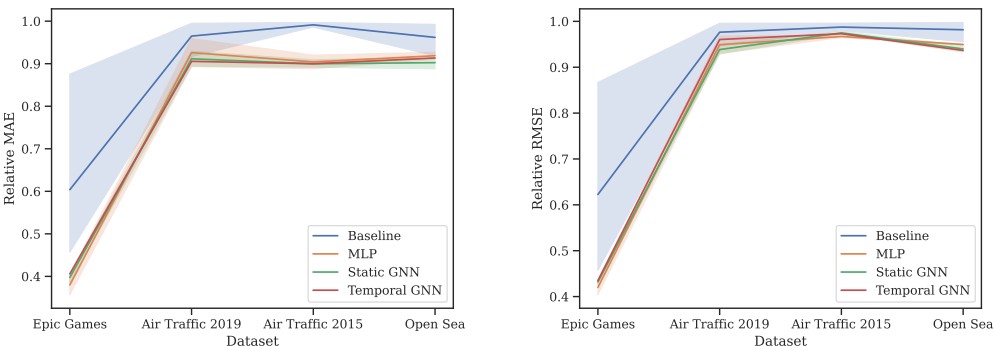

Figure 4: Relative comparison of baselines, MLP, GNN, and TGN results

The main trend that can be observed in results is that GNN-based methods generally outperform MLPs, with naive baselines performing the worst. This pattern is clearly illustrated in Figure 4. This trend of results confirms that graph information is useful to solve that task and it cannot be trivially solved by naive baselines.

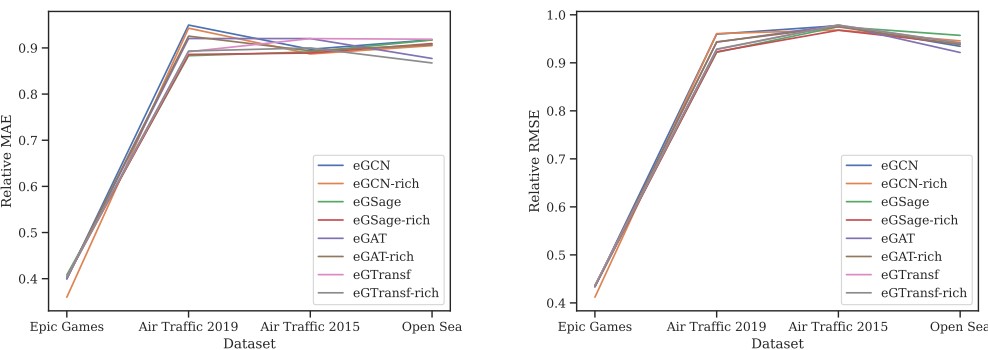

Figure 5: Relative comparison of GNNs

However, among the GNN methods, there is no clear leading method or variant that stands out; all methods appear to achieve similar performance, see Figure 5. This suggests that currently, no specific architecture based on message passing is particularly well-suited for addressing this task. This can further motivate the development of new methods tailored specifically for edge regression tasks.

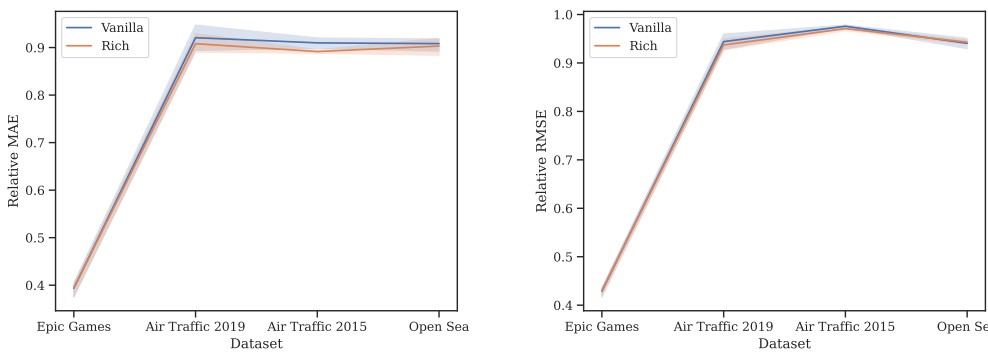

Figure 6: Relative comparison of vanilla and rich versions

Lastly, the rich variant of the GNN based methods fails to deliver significant performance improvements across all datasets, as it is highlighted in Figure 6. This reinforces the notion that existing methods are insufficient to effectively address this task, as simple adjustment are not able to provide any benefit.

**Implementation Details.** In all our experiments, data is divided into training (70%), validation (10%) and testing (20%) sets chronologically. All the models are implemented using PyTorch (Paszke et al., 2019), PyTorch Geometric (Fey and Lenssen, 2019) and PyTorch Geometric Temporal (Rozemberczki et al., 2021) libraries. All computations were run on an Nvidia DGX A100 machine with 128 AMD Rome 7742 cores and 8 Nvidia A100 GPUs.

**Hyperparameter Search.** In our experimental setup, the dimensionality of the layers in $f_{\text{conv}}(\cdot)$ is consistently set to ensure a final concatenation dimensionality of 600 before readout. The number of layers for all deep learning methods is set to 2. We conduct a grid search for the dropout probability, exploring values in $[0, 0.1, 0.3, 0.5]$. The readout function $\sigma$ is chosen dataset-dependent, with the Sigmoid function employed for Epic Games and the Tanh function for the remaining datasets. The loss function is selected from among MAE, MSE, and Huber loss. We utilize the Adam optimizer, with the learning rate tuned from a uniform distribution between 0.0001 and 0.003 and weight decay selected from $[0.0, 0.05, 0.1]$. The learnig rate scheduler is set to reduce the learning rate by a factor of 0.1 per 10, 20, or 100 steps. The batch size is set to 512 and the maximum number of epochs is set to 300, with early stopping criteria defined as no improvement in validation loss for five consecutive steps. We performed 100 steps of hyperparameters optimization to optimize the hyperparameters of all models using the software package Optuna (Akiba et al., 2019). All tuning was performed on the validation set, and we report the results on the test set that

are associated with those hyperparameter settings. The tuned values for hyperparameters are provided in the Appendix B.

## 7 Conclusions

**Summary of Contributions.** In this work, we present Temporal Edge Regression (TER) as a novel benchmark task for the temporal graph learning community. Unlike traditional graph learning tasks, TER requires forecasting continuous values, which necessitates the creation of entirely new datasets. As such, existing benchmark datasets like MOOC or REDDIT (Kumar et al., 2019) cannot be easily adapted for this purpose. To address this gap, we introduce four novel datasets specifically designed for TER. We evaluate these datasets using both simple baselines and state-of-the-art temporal graph learning methods. We believe TER closely aligns with many real-world industrial tasks, offering a level of dynamism and complexity that is often lacking in conventional graph machine learning problems. Therefore, the introduction of TER as a benchmark task represents a significant and valuable contribution to the temporal graph learning community.

**Future Work.** The introduction of TER as a benchmark task opens up several exciting avenues for future research, including but not limited to:

- **Development of Advanced Methods**: Creating more sophisticated temporal graph learning methods that are specifically tailored to the unique challenges of the TER task.

- **Exploration of Applications in Future Link Regression (FLR)**: Investigating applications and datasets that fall under the FLR setting, where none of the observations that constitute a future interaction are known until inference time.

- **Dynamic Problem Extensions**: Extending the problem formulation to more complex scenarios where (1) the set of vertices representing entities is dynamic, or (2) edges representing relationships may disappear over time, adding an additional layer of complexity and realism to the task.

## Broader Impact Statement

The study of temporal graphs has predominantly focused on tasks like dynamic node classification and future link prediction. However, these tasks represent only a fraction of the potential applications of temporal graph learning. By introducing the concept of edge regression on temporal graphs, our work broadens the scope of temporal graph analysis, opening new avenues for research and practical applications. Our research has several broader impacts:

1. Advancement in Temporal Graph Learning: By defining and formalizing the edge regression task on temporal graphs, we provide the community with a new perspective and methodology for analyzing temporal interactions between entities. This can lead to the development of more sophisticated models that better capture the dynamic nature of real-world relationships.

2. Enrichment of Benchmarking Platforms: The current landscape of open-source datasets is insufficient for thoroughly evaluating temporal edge regression models. Our introduction of four new datasets specifically designed for this task fills a critical gap in the resources available to researchers. These datasets not only enable more comprehensive benchmarking but also ensure that future models can be rigorously tested and compared.

3. Practical Applications Across Domains: Temporal edge regression has significant implications for various fields, including social network analysis, financial modeling, and recommendation systems. For instance, accurately predicting the weight or strength of future interactions (edges) can enhance the performance of recommendation algorithms, improve financial risk assessment, and provide deeper insights into social dynamics.

4. Encouraging Methodological Innovation: By evaluating a diverse set of graph learning algorithms and simple baselines, our work encourages the exploration of new methodological approaches. This can stimulate innovation and lead to the discovery of more effective techniques for temporal graph analysis.

5. Ethical Considerations and Data Use: In creating and sharing new datasets, we emphasize the importance of ethical considerations in data collection and usage. Ensuring that our datasets are anonymized and free from biases helps promote responsible research practices and the development of fair and unbiased models.

6. Educational Impact: The introduction of a new task in temporal graph learning, along with relevant datasets and baseline evaluations, provides a valuable resource for educational purposes. It can be used in academic curricula to teach students about the complexities and opportunities in temporal graph analysis, preparing the next generation of researchers and practitioners.

In summary, our work not only extends the boundaries of what is possible in temporal graph learning but also provides the necessary tools and frameworks to support future research and practical applications. Through this contribution, we aim to foster a deeper understanding and more widespread adoption of temporal graph methodologies across various domains.

## Acknowledgments and Disclosure of Funding

All funding was provided by Block Inc. [10].

---

10. block.xyz

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

## Appendix A. Preprocessing Details

### A.1 Epic Games

Each critic record is composed of following fields:

- `company` – company name that rated the game: generated an identification number that is different from any value in `game_id` for each sample in the set of unique values, used as source node.

- `author` – author comment about the game: not used because of missing values.

- `game_id` - identification of game, used as destination node.

- `date` – date of critic: converted to timestamp, used as edge time.

- `rating` – rating of game (out of 100): normalized to [0, 1], used as edge target.

- `comment` – author comment about the game: not used because observed after `date`.

- `top_critic` – verify if is a top critic (authors with verdict): not used because observed after `date`.

Each game record is composed of following fields:

- `id` – identification of game.

- `name` – name of game, `game_slug` – short name of game, and `description` –description of game: concatenated and vectorized to TF-IDF features with a vocabulary size of 512, and maximum word frequency of 0.8, used as node feature.

- `price` – price of game: normalized to [0, 1], used as node feature.

- `platform` – platforms that the game is available: converted to categorical data with 0/1 indicator.

- `genres` – genres of game: converted to categorical data with 0/1 indicator, used as node feature.

- `release_date` – release date of game: converted to timestamp, used as node feature.

- `developer` – company that developed the game: unused.

- `publisher` – company that published the game: unused.

### A.2 Air Traffic

The flight records are composed of following fields:

- `Origin` – origin IATA (International Air Transport Association) airport code: used as source node.

- `Dest` – destination IATA code: used as destination node.

- `Date` – scheduled date: used as edge time.

- `ArrTime` – actual arrival time: used to calculate edge target such that $y = $ (`ArrTime` - `CRSArrTime`) / (`CRSArrTime` - `CRSDepTime`).

- `CRSArrTime` – scheduled arrival time: used to calculate edge target such that $y = $ (`ArrTime` - `CRSArrTime`) / (`CRSArrTime` - `CRSDepTime`).

- `CRSDepTime` – scheduled departure time: used to calculate edge target such that $y = $ (`ArrTime` - `CRSArrTime`) / (`CRSArrTime` - `CRSDepTime`).

The weather conditions are summarized by following parameters for each flight:

- `dest_temperature_2m_max` – maximum daily air temperature at 2 meters above ground at destination: used as edge feature.

- `dest_temperature_2m_min` – minimum daily air temperature at 2 meters above ground at destination: used as edge feature.

- `dest_temperature_2m_mean` – mean daily air temperature at 2 meters above ground at destination: used as edge feature.

- `dest_precipitation_sum` – sum of daily precipitation at destination (including rain, showers and snowfall): used as edge feature.

- `dest_rain_sum` – sum of daily rain at destination: used as edge feature.

- `dest_snowfall_sum` – sum of daily snowfall at destination: used as edge feature.

- `dest_wind_speed_10m_max` – maximum wind gusts at destination: used as edge feature.

- `dest_wind_gusts_10m_max` – maximum wind speed at destination: used as edge feature.

- `dest_wind_direction_10m_dominant` – dominant wind direction at destination: used as edge feature.

- `origin_temperature_2m_max` – maximum daily air temperature at 2 meters above ground at origin: used as edge feature.

- `origin_temperature_2m_min` – minimum daily air temperature at 2 meters above ground at origin: used as edge feature.

- `origin_temperature_2m_mean` – mean daily air temperature at 2 meters above ground at origin: used as edge feature.

- `origin_precipitation_sum` – sum of daily precipitation at origin (including rain, showers and snowfall): used as edge feature.

- `origin_rain_sum` – sum of daily rain at origin: used as edge feature.

- **origin_snowfall_sum** – sum of daily snowfall at origin: used as edge feature.

- **origin_wind_speed_10m_max** – maximum wind gusts at origin: used as edge feature.

- **origin_wind_gusts_10m_max** – maximum wind speed at origin: used as edge feature.

- **origin_wind_direction_10m_dominant** – dominant wind direction at origin: used as edge feature.

## A.3  Open Sea

Transaction records involve following fields:

- **seller_account** – address of the NFT seller: used as source node.

- **winner_account** – address of the NFT buyer: used as destination node.

- **tx_timestamp** – timestamp of the transaction: used as edge time.

- **token** – token type used to pay the transaction: converted to categorical data with 0/1 indicator, used as edge feature.

- **chain** – blockchain where the transaction occurs: converted to categorical data with 0/1 indicator, used as edge feature.

- **token_type** – schema of the token, i.e., ERC721 or ERC1155: converted to categorical data with 0/1 indicator, used as edge feature.

- **asset_contract_type** – asset typology, i.e., non-fungible or semi-fungible: converted to categorical data with 0/1 indicator, used as edge feature.

- **asset_type** – whether the asset was involved in a simple or bundle transaction: converted to categorical data with 0/1 indicator, used as edge feature.

- **to_eth** – conversion rate to convert tokens into Ethereum at the current timestamp: normalized to [0, 1], used as edge feature.

- **to_usd** – conversion rate to convert tokens into US dollars (USD) at the current timestamp: normalized to [0, 1], used as edge feature.

- **created_date** – date of creation of the contract: converted to timestamp, used as edge feature.

- **token_id** – id of the NFT **collection_name** – id for accessing the collection name: **token_id** is unique within the same collection, so these two are used to identify unique item identification **item_id**

- **usd_price** – price of the transaction expressed in US dollars (USD): used to calculate edge target such that $y = ($usd_price$ -$ NEXT[usd_gain]$) /$ usd_price where NEXT[.] refers to subsequent transaction of **item_id**.

- **usd_gain** – difference between the price and the fees expressed in US dollars (USD): used to calculate edge target such that $y = ($usd_price$ -$ NEXT[usd_gain]$) /$ usd_price where NEXT[.] refers to subsequent transaction of **item_id**.

# Appendix B. Hyperparameter Tuning

In this section, we present the selected set of hyperparameters for each dataset and method. We performed 100 steps of hyperparameters optimization to optimize the hyperparameters of all models using the software package OPTUNA (Akiba et al., 2019). The hyperparameters were tuned on a validation set based on the best performance in terms of Mean Absolute Error (MAE).

### Table 4: Selected configurations

(a) `epic-games-plr`

|  | eMLP | eMLP-rich | eGCN | eGCN-rich | eGSage | eGSage-rich | eGAT | eGAT-rich | eGTransf | eGTransf-rich |
|---|---|---|---|---|---|---|---|---|---|---|
| batch size | 512 | 512 | 512 | 512 | 512 | 512 | 512 | 512 | 512 | 512 |
| number of epochs | 300 | 300 | 300 | 300 | 300 | 300 | 300 | 300 | 300 | 300 |
| embedding size | 600 | 600 | 600 | 600 | 600 | 600 | 600 | 600 | 600 | 600 |
| dropout | 0.5000 | 0.3000 | 0 | 0 | 0.1000 | 0.5000 | 0 | 0 | 0.1000 | 0.1000 |
| loss function | L1 | L1 | L1 | L1 | Huber | L1 | L1 | L1 | L1 | L1 |
| learning rate | 0.0001 | 0.0008 | 0.0010 | 0.0009 | 0.0012 | 0.0010 | 0.0011 | 0.0016 | 0.0023 | 0.0016 |
| weight decay | 0.0000 | 0.0500 | 0.0000 | 0.0000 | 0.1000 | 0.0500 | 0.0000 | 0.0000 | 0.0500 | 0.1000 |
| step size | 20 | 100 | 10 | 10 | 100 | 100 | 10 | 20 | 20 | 20 |
| decay factor | 0.1000 | 0.1000 | 0.1000 | 0.1000 | 0.1000 | 0.1000 | 0.1000 | 0.1000 | 0.1000 | 0.1000 |

(b) `air-traffic-2019-rlr`

|  | eMLP | eMLP-rich | eGCN | eGCN-rich | eGSage | eGSage-rich | eGAT | eGAT-rich | eGTransf | eGTransf-rich |
|---|---|---|---|---|---|---|---|---|---|---|
| batch size | 512 | 512 | 512 | 512 | 512 | 512 | 512 | 512 | 512 | 512 |
| number of epochs | 300 | 300 | 300 | 300 | 300 | 300 | 300 | 300 | 300 | 300 |
| embedding size | 600 | 600 | 600 | 600 | 600 | 600 | 600 | 600 | 600 | 600 |
| dropout | 0.1000 | 0.1000 | 0.5000 | 0 | 0.3000 | 0.3000 | 0.5000 | 0.3000 | 0.5000 | 0.3000 |
| loss function | L1 | L1 | L1 | L1 | L1 | L1 | L1 | L1 | L1 | L1 |
| learning rate | 0.0002 | 0.0024 | 0.0003 | 0.0004 | 0.0007 | 0.0017 | 0.0003 | 0.0010 | 0.0005 | 0.0005 |
| weight decay | 0.0000 | 0.0000 | 0.0000 | 0.0000 | 0.0000 | 0.0000 | 0.0000 | 0.0000 | 0.0000 | 0.0000 |
| step size | 100 | 20 | 20 | 10 | 10 | 10 | 20 | 10 | 10 | 10 |
| decay factor | 0.1000 | 0.1000 | 0.1000 | 0.1000 | 0.1000 | 0.1000 | 0.1000 | 0.1000 | 0.1000 | 0.1000 |

(c) `air-traffic-2015-rlr`

|  | eMLP | eMLP-rich | eGCN | eGCN-rich | eGSage | eGSage-rich | eGAT | eGAT-rich | eGTransf | eGTransf-rich |
|---|---|---|---|---|---|---|---|---|---|---|
| batch size | 512 | 512 | 512 | 512 | 512 | 512 | 512 | 512 | 512 | 512 |
| number of epochs | 300 | 300 | 300 | 300 | 300 | 300 | 300 | 300 | 300 | 300 |
| embedding size | 600 | 600 | 600 | 600 | 600 | 600 | 600 | 600 | 600 | 600 |
| dropout | 0.5000 | 0.3000 | 0.1000 | 0 | 0 | 0.3000 | 0.3000 | 0.1000 | 0.5000 | 0.5000 |
| loss function | L1 | L1 | L1 | L1 | L1 | L1 | L1 | L1 | Huber | L1 |
| learning rate | 0.0026 | 0.0001 | 0.0002 | 0.0004 | 0.0002 | 0.0009 | 0.0028 | 0.0007 | 0.0017 | 0.0027 |
| weight decay | 0.0000 | 0.0000 | 0.0000 | 0.0000 | 0.0000 | 0.0000 | 0.0000 | 0.0000 | 0.0000 | 0.1000 |
| step size | 100 | 100 | 10 | 10 | 10 | 10 | 20 | 10 | 10 | 10 |
| decay factor | 0.1000 | 0.1000 | 0.1000 | 0.1000 | 0.1000 | 0.1000 | 0.1000 | 0.1000 | 0.1000 | 0.1000 |

(d) `open-sea-rlr`

|  | eMLP | eMLP-rich | eGCN | eGCN-rich | eGSage | eGSage-rich | eGAT | eGAT-rich | eGTransf | eGTransf-rich |
|---|---|---|---|---|---|---|---|---|---|---|
| batch size | 512 | 512 | 512 | 512 | 512 | 512 | 512 | 512 | 512 | 512 |
| number of epochs | 300 | 300 | 300 | 300 | 300 | 300 | 300 | 300 | 300 | 300 |
| embedding size | 600 | 600 | 600 | 600 | 600 | 600 | 600 | 600 | 600 | 600 |
| dropout | 0.5000 | 0 | 0.1000 | 0.1000 | 0 | 0.3000 | 0.5000 | 0.5000 | 0.5000 | 0.1000 |
| loss function | MSE | Huber | Huber | Huber | Huber | L1 | Huber | Huber | Huber | L1 |
| learning rate | 0.0007 | 0.0001 | 0.0022 | 0.0002 | 0.0006 | 0.0003 | 0.0021 | 0.0020 | 0.0030 | 0.0002 |
| weight decay | 0.0500 | 0.1000 | 0.1000 | 0.1000 | 0.0500 | 0.1000 | 0.1000 | 0.0500 | 0.1000 | 0.1000 |
| step size | 10 | 20 | 20 | 20 | 20 | 10 | 20 | 100 | 20 | 10 |
| decay factor | 0.1000 | 0.1000 | 0.1000 | 0.1000 | 0.1000 | 0.1000 | 0.1000 | 0.1000 | 0.1000 | 0.1000 |

## Appendix C. Additional Ablation Studies

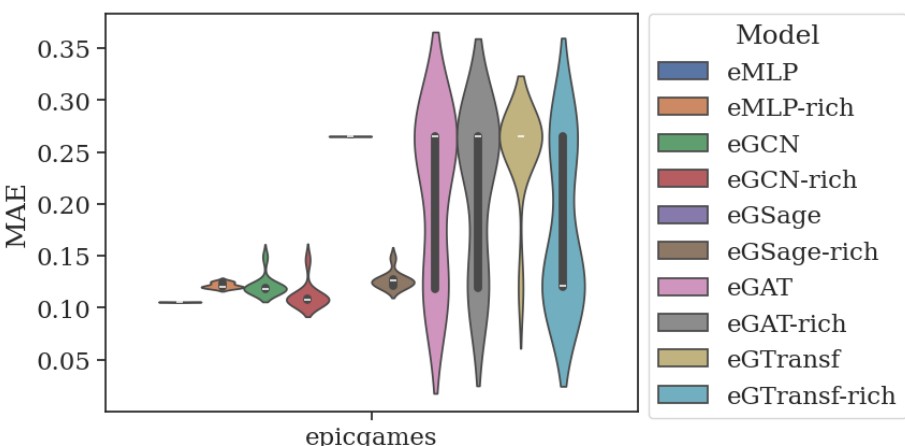

Figure 7: Kernel density estimate on MAE levels over multiple random seeds, `epic-games-plr`. Each model is run over 30 random seeds using the tuned values of hyperparameters. In general it is observed that attention based models and their variants tend to impose higher variance in performance across different random seeds.

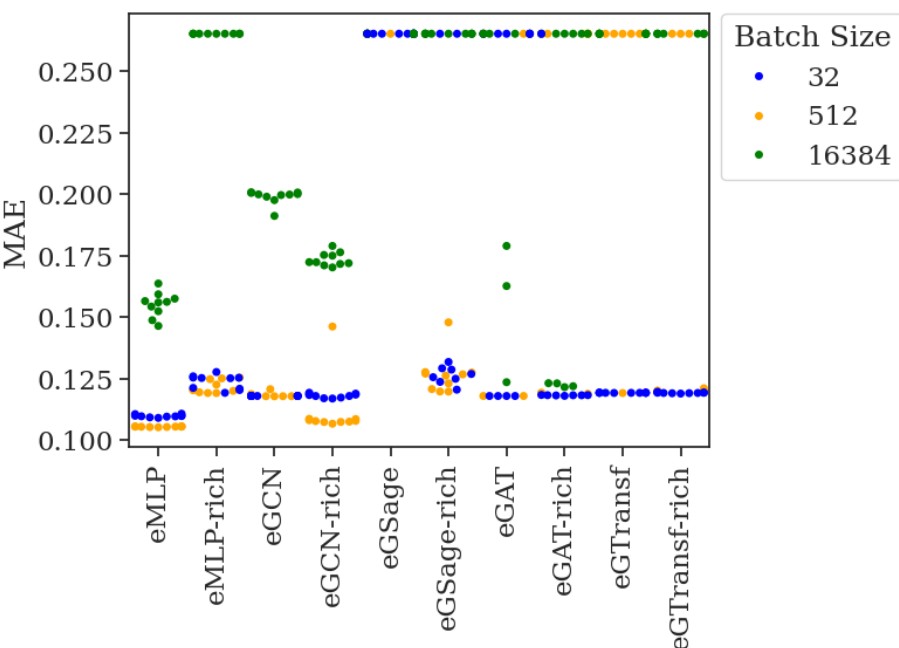

Figure 8: Ablation study for the impact of batch size, `epic-games-plr`. It is observed that as increase in batch size leads to a decrease in the performance of the model.

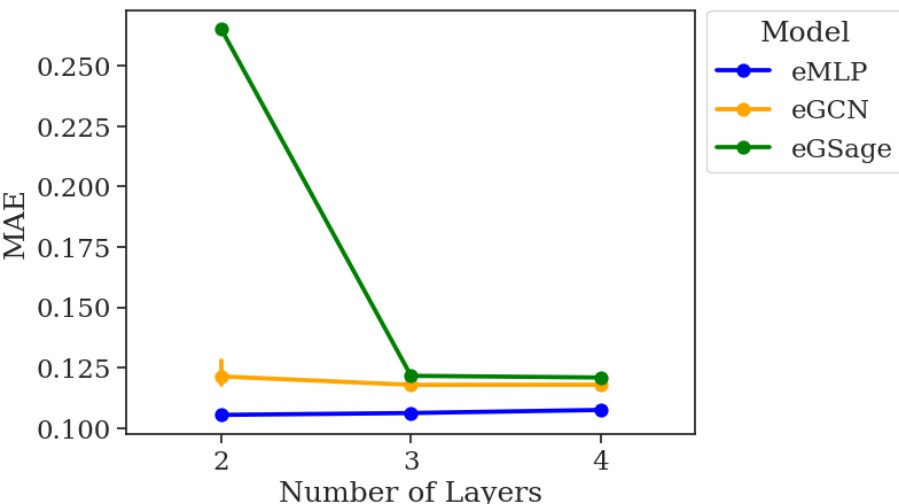

Figure 9: Ablation study for the number of layers on the `epic-games-plr`. It is observed that models deeper than 2 layers has the potential to imporve performance.

