# OpenReview forum: "Benchmarking Edge Regression on Temporal Networks"
_DMLR — Accepted by DMLR_

### Review · Reviewer_opfG · 2024-07-10

**Recommendation:** 4
**Confidence:** 2

**Summary Of Contributions:**

The paper is about temporal graph learning, which is an interesting topic. The authors four datasets that naturally lend themselves to meaningful temporal edge regression tasks. The paper is well written and well organized. However, there are several concerns in the current version of the paper that addressing them will increase the quality of this paper.

**Strengths:**

1 The datasets proposed by the authors can make a definite contribution to the graph learning community.

2 The authors give a good presentation of introducing the background.

3 The whole article has clear logic and there is nothing difficult to understand.

**Audience:**

Yes

**Claims And Evidence:**

Yes.

**Datasets And Benchmarks:**

Yes.

**Extended Submissions:**

N/A

**Limitations:**

As above.

**Requested Changes:**

1 Discuss node clustering on temporal graphs.
2 Discuss the new case.
3 Improve Table 1.
4 Give AUC performance.
5 Introduce more related works.

**Strengths And Weaknesses:**

1 The authors mainly focus on the task of link prediction on dynamic graphs. I would like to see further discussion from the authors on why these methods rarely focus on the task of node clustering.

2 Do the four datasets proposed by the authors consider the situation where two nodes have a close relationship at a certain moment, but then they cancel this relationship? How should this situation be represented in the data?

3 Table 1 does not provide enough details about the dataset. It could have included more details, such as the average degree of the nodes, the maximum and minimum number of edges between the nodes, the number of timestamps, etc.

4 For link prediction tasks, AUC is a common indicator and the corresponding results should be reported.

5 There are some new temporal graph methods that can be introduced in related work.
[1] Deep Temporal Graph Clustering. ICLR 2024.
[2] Hypergraph Dynamic System. ICLR 2024.

---

### Review · Reviewer_uLLd · 2024-07-30

**Recommendation:** 3
**Confidence:** 2

**Summary Of Contributions:**

1. **Introduction of Temporal Edge Regression (TER):** The paper introduces the task of Temporal Edge Regression, which aims to predict a continuous target value associated with an edge based on past observations and time-varying graph connectivity. This task extends the existing work on dynamic node classification and link prediction.


2. **Proposal of Four Novel Datasets:** The authors introduce four new datasets that are suitable for TER task.

4. **Benchmarking and Evaluation:** The paper benchmarks the performance of several methods, including non-parametric methods, NN-based ones.

5. **Problem Definition**The authors define the problem of TER and its variations based on the timing of observations, including RLR, PLR and FLR.

**Strengths:**

1. This paper is clearly written and well-organized. As a dataset/benchmark paper, the authors have provided comprehensive details on the problem definition, dataset construction, and baseline comparisons.

2. The authors not only clearly define temporal link regression but also succinctly categorize it into three subtypes (RLR, PLR, and FLR), providing corresponding datasets for each subtype.

**Audience:**

Yes

**Broader Impact Concerns:**

I have no concerns about the ethical implications of the work.

**Claims And Evidence:**

Yes

**Datasets And Benchmarks:**

The authors did not provide sufficient detail on:
1. data organization. For example, how they process the textual information;
2. data availability. Currently, the dataset repository they provide (https://huggingface.co/datasets/ca-aird) is not accessible;
3. data maintenance.

**Extended Submissions:**

No, this is not an extended version.

**Limitations:**

1. The authors should provide more details on graph construction. For example, in Section 4.1 on Epic Games, they mention, "The features of game vertices are extracted from textual data such as game descriptions, nominal data such as genres, and interval data such as price." Specifically, how is the textual information transformed into node features? Did you use TF-IDF+PCA? What is the total dimensionality of the node features, and what does each dimension represent? The authors can refer to similar papers like [1,2].

2. The selection of temporal/dynamic GNN backbones by the authors is insufficient. There are now much more advanced temporal GNN methods available than TGN. I strongly recommend the authors refer to [3].

3. The experiments are highly insufficient. Additional experiments the authors should conduct include, but are not limited to: sensitivity analysis of GNN models on the four datasets regarding the number of GNN layers and batch size, sensitivity analysis of temporal GNNs to sliding window size, and GPU memory usage for different methods.


[1] Anti-Money Laundering in Bitcoin: Experimenting with Graph Convolutional Networks for Financial Forensics, KDD'19

[2] Collective Opinion Spam Detection: Bridging Review Networks and Metadata， KDD'15

[3] A Comprehensive Survey of Dynamic Graph Neural Networks: Models, Frameworks, Benchmarks, Experiments and Challenges

**Requested Changes:**

(Items with * indicate proposed adjustments that are critical to securing my recommendation for acceptance, while others would simply strengthen the work itself in my view.)

1. *The authors should report more dataset statistics in Table 1, such as node feature dimension, edge feature dimension, time range, etc. Please refer to Table 2 in [1].
2. *In Figure 3, both (b) and (c) represent air traffic data, yet their edge homophily trends differ significantly. The authors should explain the reasons for the increase or decrease in homophily trends and why there is a difference between (b) and (c).
3. *The authors need to test their datasets using additional recent temporal/dynamic GNN backbones.
4. The authors use [2] for their eGraphTransformer implementation. I would personally recommend GraphGPS[3], as it is now widely recognized as a benchmark for graph transformers with more reliable performance.
5. *The authors should include more sensitivity analysis, as suggested in Limitations (3).
6. When constructing the two air traffic datasets, the authors supplemented the flight datasets with weather data from Open-Meteo. I am curious whether the integration of these external features genuinely enhanced the dataset's capacity. The authors might consider adding corresponding ablation experiments.

[1] A Comprehensive Survey of Dynamic Graph Neural Networks: Models, Frameworks, Benchmarks, Experiments and Challenges

[2] Graph transformer networks, NeurIPS'19

[3] Recipe for a General, Powerful, Scalable Graph Transformer, NeurIPS'22

**Strengths And Weaknesses:**

Please refer to `Strengths` and `Limitations`.

---

### Review · Reviewer_qRhE · 2024-08-01

**Recommendation:** 3
**Confidence:** 2

**Summary Of Contributions:**

1. New Task Definition: Proposing and defining the TER task to predict continuous target values for edges based on past observations and time-varying graph connectivity.
2. New Datasets: Introducing four new datasets (epic-games-plr, air-traffic-2015-rlr, air-traffic-2019-rlr, open-sea-rlr) that are specifically designed for meaningful temporal edge regression tasks.
3. Benchmarking: Evaluating a set of methods, including popular graph learning algorithms and simple baselines like vertex-based moving average, to establish benchmarks for the TER task.

**Strengths:**

See above.

**Audience:**

Yes

**Broader Impact Concerns:**

Do not have broader impact concerns.

**Claims And Evidence:**

The claims made in the submission are supported by accurate, convincing and clear evidence.

**Datasets And Benchmarks:**

For datasets, there is sufficient detail on data collection and organization, availability and maintenance, and ethical and responsible use. But the authors may increase the diversity of datasets.
For benchmarks, there is sufficient detail to support reproducibility. However, the authors can include more temporal graph models for comparison.

**Extended Submissions:**

To the best of my knowledge, the submission is not the extended version of a previously published work.

**Limitations:**

Additional Method Comparisons:
Including comparisons with additional baseline methods, especially those that work on recent link prediction on dynamic graph, would provide a more comprehensive benchmarking and context for the performance of different methods.
Inclusion of Future Link Regression (FLR) Tasks:
Expanding the study to include FLR tasks would provide a more comprehensive understanding of Temporal Edge Regression (TER).

**Requested Changes:**

See weaknesses above.

**Strengths And Weaknesses:**

Strong Aspects:
1. Innovative and Meaningful Task: The introduction of temporal edge regression (TER) fills a gap in the existing temporal graph learning tasks, which have been limited to dynamic node classification and future and recent link prediction.
2. Comprehensive Evaluation: The paper provides a thorough evaluation of different methods on the newly proposed datasets, offering valuable insights into the performance of these methods on TER tasks.
3. Dataset Availability: Making the processed versions of the proposed datasets accessible to the research community enhances reproducibility and encourages further research in this area.

Weaker Elements:
1.The four datasets introduced in this paper all have fixed node sets, but there are many other temporal graph datasets with very different statistics, although they are not designed for edge regression. But it’s ok since this is the first paper for temporal edge regression.
2.I agree that the rich variants of the GNN-based methods did not show significant performance improvements across the datasets, which might indicate that additional features or more complex models are necessary for better performance. However, the paper only used one temporal graph method TGN to perform experiments. Maybe the authors can try more temporal graph methods to see how they perform on the TER task. Otherwise, it’s hard to make a conclusion that current temporal graph methods are not well-suited for TER task.

---

### Review · Reviewer_156K · 2024-08-01

**Recommendation:** 4
**Confidence:** 3

**Summary Of Contributions:**

This paper introduces the novel problem of temporal edge regression (TER) on dynamic graphs and proposes four datasets tailored for this task. It evaluates various baseline methods, including moving average, edge similarity, graph neural networks and temporal graph network, on the proposed datasets. The paper makes a valuable contribution by defining a new benchmark problem and releasing usable datasets to motivate further research.

**Strengths:**

- It clearly defines the new problem of temporal edge regression on dynamic graphs and provides a solid problem formulation.
- It constructs four novel datasets from real-world applications that are suitable for evaluating temporal edge regression, including game reviews, flight delays, and NFT transactions. The datasets will enable further research on this task.
- It provides baseline results from various methods to quantitatively evaluate temporal edge regression and analyze the effectiveness of exploiting graph structure, which can guide future model development.

**Audience:**

Yes

**Broader Impact Concerns:**

The paper introduces new data and methods for predictive modeling but does not discuss their broader societal implications. Some concerns that may warrant consideration in a Broader Impact statement:
- Privacy and consent: The datasets contain sensitive transaction/interaction data but no information is provided on how individuals' privacy and consent were ensured.
- Disparate harms: Predictions made by these models, especially for high-risk applications, could differentially impact vulnerable groups if biases are not mitigated.
- Accountability: As an open benchmark, these models and data could be adopted widely without oversight on how they are employed or potential harms addressed. Clarifying responsible development and use standards could help maximize benefits and prevent misuse.

**Claims And Evidence:**

Yes

**Datasets And Benchmarks:**

The paper provides good levels of detail on the datasets proposed for the new temporal edge regression (TER) task:
- Collection and organization details are provided for each dataset, including descriptions of the raw data sources (Epic games, air travel data, OpenSea transactions), how the graph structures were constructed from these sources, and how node/edge features and target variables were calculated.
- preprocessing steps taken are documented in appendices for each dataset. This would support reproducibility.

**Extended Submissions:**

No, this is not an extended version.

**Limitations:**

Qualitative Evaluation:
As with any large-scale system, biases may emerge from the data or algorithms used for tasks like paper matching, reviewer assignment, or content moderation. The authors have not discussed potential biases or measures taken to ensure fairness.
Generalizability:
Only four datasets are introduced for the new temporal edge regression task. Evaluating on more diverse datasets from other domains would help assess how well the conclusions generalize.

**Requested Changes:**

This paper introduces an important new problem of temporal edge regression and provides useful resources to advance research. However, some adjustments would be needed to fully evaluate the conclusions and secure my recommendation for acceptance.

Critical Changes:
- Provide more implementation details for reproducibility. Specify model architectures, hyperparameters, training procedures etc. for each method using pseudocode or diagram. This is critical to assess the results.
- Perform statistical testing to determine significance of performance differences. Use welch's t-test or similar and report p-values. It is critical to confirm conclusions rather than rely on ranking.
- Conduct ablation studies to analyze impact of model components like temporal encoding, edge features etc. Remove each in turn and measure effect on MAE/RMSE scores. This is needed to establish value of specific design decisions.

Strengthening Changes:
- Consider additional temporal graph learning baselines like TGN variants or GRU-based methods to allow more comprehensive evaluation.
- Characterize graph dynamics over time by examining statistics like $ |E(t)|, |V(t)| $. This provides context to interpret results.
- Study temporal homophily variations rather than aggregate score. Correlations between $ H_e(t) $. and performance may yield insights.

**Strengths And Weaknesses:**

Paper Strengths:
- It clearly defines the new problem of temporal edge regression on dynamic graphs and provides a solid problem formulation.
- It constructs four novel datasets from real-world applications that are suitable for evaluating temporal edge regression, including game reviews, flight delays, and NFT transactions. The datasets will enable further research on this task.
- It provides baseline results from various methods to quantitatively evaluate temporal edge regression and analyze the effectiveness of exploiting graph structure, which can guide future model development.


Paper Weaknesses:
- More advanced temporal graph learning methods could be evaluated as baselines for a more comprehensive comparison.
- The edge homophily analysis only considers aggregate homophily over time instead of studying its variation. Investigating temporal dynamics of homophily could provide more insights.
- For temporal edge regression methods like TGN, the memory updating scheme is not clearly explained. More implementation details are needed to understand how historical information is captured.
- Potential overfitting due to imbalanced training/validation splits is not addressed. Evaluation on hidden tests would strengthen conclusions.